# FINE-GRAINED SYNTHESIS OF UNRESTRICTED ADVERSARIAL EXAMPLES

## ABSTRACT

We propose a novel approach for generating unrestricted adversarial examples by manipulating fine-grained aspects of image generation. Unlike existing unrestricted attacks that typically hand-craft geometric transformations, we learn stylistic and stochastic modifications leveraging state-of-the-art generative models. This allows us to manipulate an image in a controlled, fine-grained manner without being bounded by a norm threshold. Our approach can be used for targeted and non-targeted unrestricted attacks on classification, semantic segmentation and object detection models. Our attacks can bypass certified defenses, yet our adversarial images look indistinguishable from natural images as verified by human evaluation. Moreover, we demonstrate that adversarial training with our examples *improves* performance of the model on clean images without requiring any modifications to the architecture. We perform experiments on LSUN, CelebA-HQ and COCO-Stuff as high resolution datasets to validate efficacy of our proposed approach.

## 1 INTRODUCTION

Adversarial examples, inputs resembling real samples but maliciously crafted to mislead machine learning models, have been studied extensively in the last few years. Most of the existing papers, however, focus on norm-constrained attacks and defenses, in which the adversarial input lies in an $\epsilon$-neighborhood of a real sample using the $L_p$ distance metric (commonly with $p = 0, 2, \infty$). For small $\epsilon$, the adversarial input is quasi-indistinguishable from the natural sample. For an adversarial image to fool the human visual system, it is sufficient to be norm-constrained; but this condition is not necessary. Moreover, defenses tailored for norm-constrained attacks can fail on other subtle input modifications. This has led to a recent surge of interest on unrestricted adversarial attacks in which the adversary is not bounded by a norm threshold. These methods typically hand-craft transformations to capture visual similarity. Spatial transformations [Engstrom et al. (2017); Xiao et al. (2018); Alaifari et al. (2018)], viewpoint or pose changes [Alcorn et al. (2018)], inserting small patches [Brown et al. (2017)], among other methods, have been proposed for unrestricted adversarial attacks.

In this paper, we focus on fine-grained manipulation of images for unrestricted adversarial attacks. We build upon state-of-the-art generative models which disentangle factors of variation in images. We create fine and coarse-grained adversarial changes by manipulating various latent variables at different resolutions. Loss of the target network is used to guide the generation process. The pre-trained generative model constrains the search space for our adversarial examples to realistic images, thereby revealing the target model's vulnerability in the natural image space. We verify that we do not deviate from the space of realistic images with a user study as well as a t-SNE plot comparing distributions of real and adversarial images (see Fig. 7 in the appendix). As a result, we observe that including these examples in training the model enhances its accuracy on clean images.

Our contributions can be summarized as follows:

- We present the first method for *fine-grained* generation of high-resolution unrestricted adversarial examples in which the attacker controls which aspects of the image to manipulate, resulting in a diverse set of realistic, on-the-manifold adversarial examples.

- We demonstrate that adversarial training with our examples *improves* performance of the model on clean images. This is in contrast to training with norm-bounded perturbations which *degrades* the model's accuracy. Unlike recent approaches such as Xie et al. (2020) which use a separate auxiliary batch norm for adversarial examples, our method does not require any modifications to the architecture.

- We propose the first method for generating unrestricted adversarial examples for semantic segmentation and object detection. Training with our examples improves segmentation results on clean images.

- We demonstrate that our proposed attack can break certified defenses on norm-bounded perturbations.

## 2 RELATED WORK

### 2.1 NORM-CONSTRAINED ADVERSARIAL EXAMPLES

Most of the existing works on adversarial attacks and defenses focus on norm-constrained adversarial examples: for a given classifier $F : \mathbb{R}^n \rightarrow \{1, \ldots, K\}$ and an image $x \in \mathbb{R}^n$, the adversarial image $x' \in \mathbb{R}^n$ is created such that $\|x - x'\|_p < \epsilon$ and $F(x) \neq F(x')$. Common values for $p$ are $0, 2, \infty$, and $\epsilon$ is chosen small enough so that the perturbation is imperceptible. Various algorithms have been proposed for creating $x'$ from $x$. Optimization-based methods solve a surrogate optimization problem based on the classifier's loss and the perturbation norm. In their pioneering paper on adversarial examples, Szegedy et al. (2013) use box-constrained L-BFGS [Fletcher (2013)] to minimize the surrogate loss function. Carlini & Wagner (2017) propose stronger optimization-based attacks for $L_0, L_2$ and $L_\infty$ norms using better objective functions and the Adam optimizer. Gradient-based methods use gradient of the classifier's loss with respect to the input image. Fast Gradient Sign Method (FGSM) [Goodfellow et al. (2014)] uses a first-order approximation of the function for faster generation and is optimized for the $L_\infty$ norm. Projected Gradient Descent (PGD) [Madry et al. (2017)] is an iterative variant of FGSM which provides a strong first-order attack by using multiple steps of gradient ascent and projecting perturbed images to an $\epsilon$-ball centered at the input. Other variants of FGSM are proposed by Dong et al. (2018) and Kurakin et al. (2016).

Several methods have been proposed for defending against adversarial attacks. These approaches can be broadly categorized to *empirical* defenses which are empirically robust to adversarial examples, and *certified* defenses which are provably robust to a certain class of attacks. One of the most successful empirical defenses is *adversarial training* [Goodfellow et al. (2014); Kurakin et al. (2016); Madry et al. (2017)] which augments training data with adversarial examples generated as the training progresses. Many empirical defenses attempt to combat adversaries using a form of input pre-processing or by manipulating intermediate features or gradients [Guo et al. (2017); Xie et al. (2017); Samangouei et al. (2018)]. Few approaches have been able to scale up to high-resolution datasets such as ImageNet [Liao et al. (2018); Xie et al. (2018); Kannan et al. (2018)]. Athalye et al. (2018) show that many of these defenses fail due to *obfuscated gradients*, which occurs when the defense method is designed to mask information about the model's gradients. Vulnerabilities of empirical defenses have led to increased interest in certified defenses, which provide a guarantee that the classifier's prediction is constant within a neighborhood of the input. Several certified defenses have been proposed [Wong & Kolter (2017); Raghunathan et al. (2018); Tsuzuku et al. (2018)] which typically do not scale to ImageNet. Cohen et al. (2019) use randomized smoothing with Gaussian noise to obtain provably $L_2$-robust classifiers on ImageNet. Lecuyer et al. (2019) propose an alternative certified defense at ImageNet scale leveraging a connection between robustness against adversarial examples and differential privacy theory.

### 2.2 UNRESTRICTED ADVERSARIAL EXAMPLES

For an image to be adversarial, it needs to be visually indistinguishable from real images. One way to achieve this is by applying subtle geometric transformations to the input image. Spatially transformed adversarial examples are introduced by Xiao et al. (2018) in which a flow field is learned to displace pixels of the image. Similarly, Alaifari et al. (2018) iteratively apply small deformations to the input in order to obtain the adversarial image. Engstrom et al. (2017) show that simple translations and rotations are enough for fooling deep neural networks. Alcorn et al. (2018) manipulate pose of an object to fool deep neural networks. They estimate parameters of a 3D renderer that cause the target model to misbehave in response to the rendered image. Another approach for evading the norm constraint is to insert new objects in the image. Adversarial Patch [Brown et al. (2017)] creates an adversarial image by completely replacing part of an image with a synthetic patch, which is image-agnostic and robust to transformations. Existence of on-the-manifold adversarial examples is also shown by Gilmer et al. (2018), that consider the task of classifying between two concentric n-dimensional spheres. Stutz et al. (2019) demonstrate that both robust and accurate models are possible by using on-the-manifold adversarial examples. A challenge for creating unrestricted adversarial examples and defending against them is introduced by Brown et al. (2018) using the simple task of classifying between birds and bicycles. The recent work by Gowal et al. (2020) show that adversarial training with examples generated by StyleGAN can improve performance of the model on clean images. They consider the classification task on low-resolution datasets such as ColorMNIST and CelebA, and only use fine changes in their adversarial training. Our approach is effective on high-resolution datasets such as CelebA-HQ and LSUN, uses a range of low-level to high-level changes for adversarial training and encompasses several tasks including classification, segmentation and detection. In addition, we demonstrate that our adversarial examples can break certified defenses on norm-constrained perturbations and are realistic as verified by human evaluation. Song

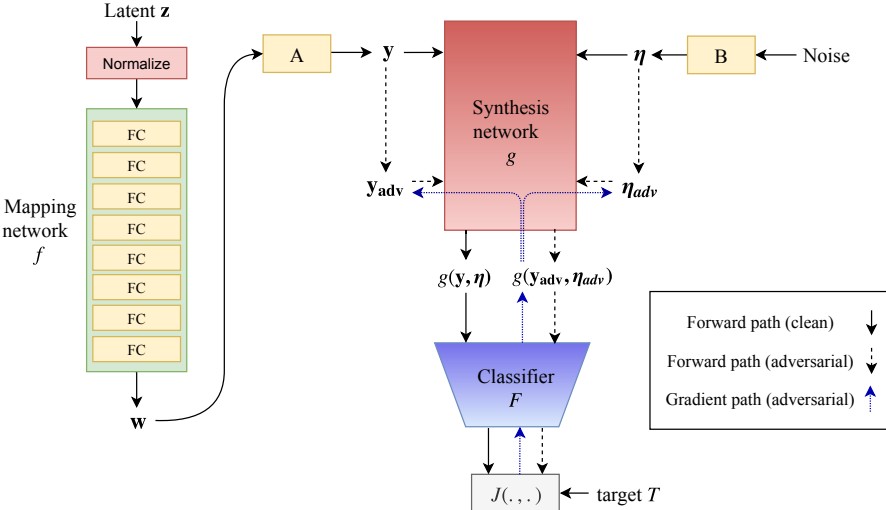

Figure 1: Classification architecture. Style ($\mathbf{y}$) and noise ($\boldsymbol{\eta}$) variables are used to generate images $g(\mathbf{y}, \boldsymbol{\eta})$ which are fed to the classifier $F$. Adversarial style and noise tensors are initialized with $\mathbf{y}$ and $\boldsymbol{\eta}$ and iteratively updated using gradients of the loss function $J$.

et al. (2018) search in the latent ($z$) space of AC-GAN [Odena et al. (2017)] to find generated images that can fool a target classifier but yield correct predictions on AC-GAN's auxiliary classifier. They constrain the search region of $z$ so that it is close to a randomly sampled noise vector, and show results on MNIST, SVHN and CelebA datasets. Requiring two classifiers to have inconsistent predictions degrades sample quality of the model. As we show in the appendix, training with these adversarial examples hurts the model's performance on clean images. Moreover, this approach has no control over the generation process since small changes in the $z$ space can lead to large changes in generated images and even create unrealistic samples. On the other hand, our method manipulates high-resolution real or synthesized images in a fine-grained manner owing to the interpretable disentangled latent space. It also generates samples which improve the model's accuracy on clean images both in classification and segmentation tasks. To further illustrate difference of our approach with Song et al. (2018), we plot t-SNE embeddings of real images from CelebA-HQ as well as adversarial examples from our method and Song et al.'s approach in the appendix and show that our adversarial images stay closer to the manifold of real images.

## 3 APPROACH

Most of the existing works on unrestricted adversarial attacks rely on geometric transformations and deformations which are oblivious to latent factors of variation. In this paper, we leverage disentangled latent representations of images for unrestricted adversarial attacks. We build upon state-of-the-art generative models and consider various target tasks: classification, semantic segmentation and object detection.

### 3.1 CLASSIFICATION

Style-GAN [Karras et al. (2018)] is a state-of-the-art generative model which disentangles high-level attributes and stochastic variations in an unsupervised manner. Stylistic variations are represented by *style* variables and stochastic details are captured by *noise* variables. Changing the noise only affects low-level details, leaving the overall composition and high-level aspects intact. This allows us to manipulate the noise variables such that variations are barely noticeable by the human eye. The style variables affect higher level aspects of image generation. For instance, when the model is trained on bedrooms, style variables from the top layers control viewpoint of the camera, middle layers select the particular furniture, and bottom layers deal with colors and details of materials. This allows us to manipulate images in a controlled manner, providing an avenue for fine-grained unrestricted attacks.

Formally, we can represent Style-GAN with a mapping function $f$ and a synthesis network $g$. The mapping function is an 8-layer MLP which takes a latent code $\mathbf{z}$, and produces an intermediate latent vector $\mathbf{w} = f(\mathbf{z})$. This vector is then specialized by learned affine transformations $A$ to style variables $\mathbf{y}$, which control adaptive instance normalization operations after each convolutional layer of the synthesis network $g$. Noise inputs are single-channel images consisting of un-correlated Gaussian noise that are fed to each layer of the synthesis network. Learned per-feature scaling factors $B$ are used to generate noise variables $\boldsymbol{\eta}$ which are added to the output of convolutional

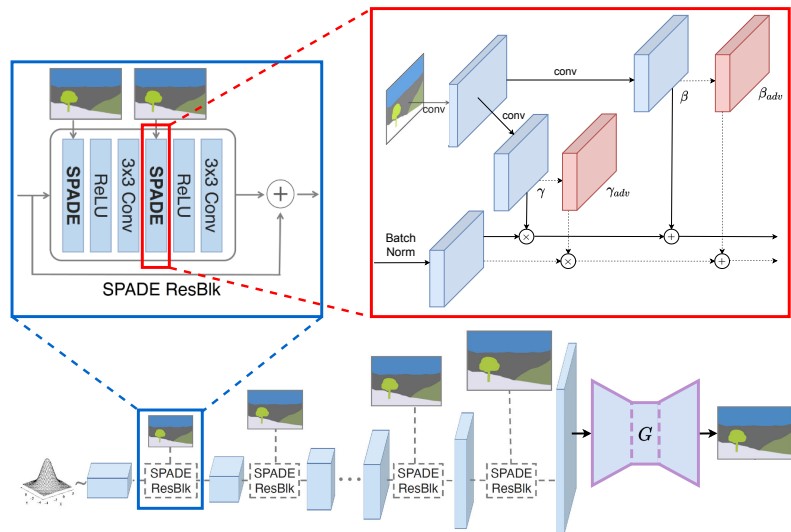

Figure 2: Semantic segmentation architecture. Adversarial parameters $\gamma_{adv}$ and $\beta_{adv}$ are initialized with $\gamma$ and $\beta$, and iteratively updated to fool the segmentation model $G$.

layers. The synthesis network takes style $\mathbf{y}$ and noise $\boldsymbol{\eta}$ as input, and generates an image $\mathbf{x} = g(\mathbf{y}, \boldsymbol{\eta})$. We pass the generated image to a pre-trained classifier $F$. We seek to slightly modify $\mathbf{x}$ so that $F$ can no longer classify it correctly. We achieve this through perturbing the style and noise tensors. We initialize adversarial style and noise variables as $\mathbf{y}_{\mathbf{adv}}^{(\mathbf{0})} = \mathbf{y}$ and $\boldsymbol{\eta}_{\mathbf{adv}}^{(\mathbf{0})} = \boldsymbol{\eta}$, and iteratively update them in order to fool the classifier. Loss of the classifier determines the update rule, which in turn depends on the type of attack. As common in the literature, we consider two types of attacks: non-targeted and targeted.

In order to generate non-targeted adversarial examples, we need to change the model's original prediction. Starting from initial values $\mathbf{y}_{\mathbf{adv}}^{(\mathbf{0})} = \mathbf{y}$ and $\boldsymbol{\eta}_{\mathbf{adv}}^{(\mathbf{0})} = \boldsymbol{\eta}$, we can iteratively perform gradient ascent in the style and noise spaces of the generator to find values that maximize the classifier's loss. Alternatively, as proposed by Kurakin et al. (2016), we can use the least-likely predicted class $ll_{\mathbf{x}} = \arg\min(F(\mathbf{x}))$ as our target. We found this approach more effective in practice. At time step $t$, the update rule for the style and noise variables is:

$$\mathbf{y}_{\mathbf{adv}}^{(\mathbf{t+1})} = \mathbf{y}_{\mathbf{adv}}^{(\mathbf{t})} - \epsilon \cdot \mathrm{sign}(\nabla_{\mathbf{y}_{\mathbf{adv}}^{(\mathbf{t})}} J(F(g(\mathbf{y}_{\mathbf{adv}}^{(\mathbf{t})}, \boldsymbol{\eta}_{\mathbf{adv}}^{(\mathbf{t})})), ll_{\mathbf{x}})) \tag{1}$$

$$\boldsymbol{\eta}_{\mathbf{adv}}^{(\mathbf{t+1})} = \boldsymbol{\eta}_{\mathbf{adv}}^{(\mathbf{t})} - \delta \cdot \mathrm{sign}(\nabla_{\boldsymbol{\eta}_{\mathbf{adv}}^{(\mathbf{t})}} J(F(g(\mathbf{y}_{\mathbf{adv}}^{(\mathbf{t})}, \boldsymbol{\eta}_{\mathbf{adv}}^{(\mathbf{t})})), ll_{\mathbf{x}})) \tag{2}$$

in which $J(\cdot, \cdot)$ is the classifier's loss function, $F(\cdot)$ gives the probability distribution over classes, $\mathbf{x} = g(\mathbf{y}, \boldsymbol{\eta})$, and $\epsilon, \delta \in \mathbb{R}$ are step sizes. We use $(\epsilon, \delta) = (0.004, 0.2)$ and $(0.004, 0.1)$ for LSUN and CelebA-HQ respectively. We perform multiple steps of gradient descent (usually 2 to 10) until the classifier is fooled.

Generating targeted adversarial examples is more challenging as we need to change the prediction to a specific class $T$. In this case, we perform gradient descent to minimize the classifier's loss with respect to the target:

$$\mathbf{y}_{\mathbf{adv}}^{(\mathbf{t+1})} = \mathbf{y}_{\mathbf{adv}}^{(\mathbf{t})} - \epsilon \cdot \mathrm{sign}(\nabla_{\mathbf{y}_{\mathbf{adv}}^{(\mathbf{t})}} J(F(g(\mathbf{y}_{\mathbf{adv}}^{(\mathbf{t})}, \boldsymbol{\eta}_{\mathbf{adv}}^{(\mathbf{t})})), T)) \tag{3}$$

$$\boldsymbol{\eta}_{\mathbf{adv}}^{(\mathbf{t+1})} = \boldsymbol{\eta}_{\mathbf{adv}}^{(\mathbf{t})} - \delta \cdot \mathrm{sign}(\nabla_{\boldsymbol{\eta}_{\mathbf{adv}}^{(\mathbf{t})}} J(F(g(\mathbf{y}_{\mathbf{adv}}^{(\mathbf{t})}, \boldsymbol{\eta}_{\mathbf{adv}}^{(\mathbf{t})})), T)) \tag{4}$$

We use $(\epsilon, \delta) = (0.005, 0.2)$ and $(0.004, 0.1)$ in the experiments on LSUN and CelebA-HQ respectively. In practice 3 to 15 updates suffice to fool the classifier. Note that we only control deviation from the initial latent variables, and do not impose any norm constraint on generated images.

### 3.1.1 INPUT-CONDITIONED GENERATION

Generation can also be conditioned on real input images by embedding them into the latent space of Style-GAN. We first synthesize images similar to the given input image $I$ by optimizing values of $\mathbf{y}$ and $\boldsymbol{\eta}$ such that $g(\mathbf{y}, \boldsymbol{\eta})$ is close to $I$. More specifically, we minimize the perceptual distance [Johnson et al. (2016)] between $g(\mathbf{y}, \boldsymbol{\eta})$ and $I$. We can then proceed similar to equations 1–4 to perturb these tensors and generate the adversarial image. Realism of synthesized images depends on inference properties of the generative model. In practice, generated images resemble input images with high fidelity especially for CelebA-HQ images.

### 3.2 Semantic Segmentation and Object Detection

We also consider the task of semantic segmentation and leverage the generative model proposed by Park et al. (2019). The model is conditioned on input semantic layouts and uses SPatially-Adaptive (DE)normalization (SPADE) modules to better preserve semantic information against common normalization layers. The layout is first projected onto an embedding space and then convolved to produce the modulation parameters $\gamma$ and $\beta$. We adversarially modify these parameters with the goal of fooling a segmentation model. We consider non-targeted attacks using per-pixel predictions and compute gradient of the loss function with respect to the modulation parameters with an update rule similar to equations 1 and 2. Figure 2 illustrates the architecture. Note that manipulating variables at smaller resolutions lead to coarser changes. We consider a similar architecture for the object detection task except that we pass the generated image to the detection model and try to increase its loss. Results for this task are shown in the appendix.

## 4 Results and Discussion

We provide qualitative and quantitative results using experiments on LSUN [Yu et al. (2015)] and CelebA-HQ [Karras et al. (2017)]. LSUN contains 10 scene categories and 20 object categories. We use all the scene classes as well as two object classes: *cars* and *cats*. We consider this dataset since it is used in Style-GAN, and is well suited for a classification task. For the scene categories, a 10-way classifier is trained based on Inception-v3 [Szegedy et al. (2016)] which achieves an accuracy of $87.7\%$ on LSUN's test set. The two object classes also appear in ImageNet [Deng et al. (2009)], a richer dataset containing 1000 categories. Therefore, for experiments on cars and cats we use an Inception-v3 model trained on ImageNet. This allows us to explore a broader set of categories in our attacks, and is particularly helpful for targeted adversarial examples. CelebA-HQ consists of 30,000 face images at $1024 \times 1024$ resolution. We consider the gender classification task, and use the classifier provided by Karras et al. (2018). This is a binary task for which targeted and non-targeted attacks are similar.

In order to synthesize a variety of adversarial examples, we use different random seeds in Style-GAN to obtain various values for $\mathbf{z}, \mathbf{w}, \mathbf{y}$ and $\boldsymbol{\eta}$. Style-based adversarial examples are generated by initializing $\mathbf{y_{adv}}$ with the value of $\mathbf{y}$, and iteratively updating it as in equation 1 (or 3) until the resulting image $g(\mathbf{y_{adv}}, \boldsymbol{\eta})$ fools the classifier $F$. Noise-based adversarial examples are created similarly using $\boldsymbol{\eta_{adv}}$ and the update rule in equation 2 (or 4). While using different step sizes makes a fair comparison difficult, we generally found it easier to fool the model by manipulating the noise variables. We can also combine the effect of style and noise by simultaneously updating $\mathbf{y_{adv}}$ and $\boldsymbol{\eta_{adv}}$ in each iteration, and feeding $g(\mathbf{y_{adv}}, \boldsymbol{\eta_{adv}})$ to the classifier. In this case, the effect of style usually dominates since it creates coarser changes.

Figure 3 illustrates generated adversarial examples on LSUN. Original image $g(\mathbf{y}, \boldsymbol{\eta})$, noise-based image $g(\mathbf{y}, \boldsymbol{\eta_{adv}})$ and style-based image $g(\mathbf{y_{adv}}, \boldsymbol{\eta})$ are shown. Adversarial images look almost indistinguishable from natural images. Manipulating the noise variable results in subtle, imperceptible changes. Varying the style leads to coarser changes such as different colorization, pose changes, and even removing or inserting objects in the scene. We can also control granularity of changes by selecting specific layers of the model. Manipulating top layers, corresponding to coarse spatial resolutions, results in high-level changes. Lower layers, on the other hand, modify finer details. In the first two columns of Figure 3, we only modify top 6 layers (out of 18) to generate adversarial images. The middle two columns change layers 7 to 12, and the last column uses the bottom 6 layers.

Figure 4 depicts adversarial examples on CelebA-HQ gender classification. Males are classified as females and vice versa. As we observe, various facial features are altered by the model yet the identity is preserved. Similar to LSUN images, noise-based changes are more subtle than style-based ones, and we observe a spectrum of high-level, mid-level and low-level changes. Figure 5 illustrates adversarial examples conditioned on real input images using the procedure described in Section 3.1.1. Synthesized images resemble inputs with high fidelity, and set the initial values in our optimization process. In some cases, we can notice how the model is altering masculine or feminine features. For instance, women's faces become more masculine in columns 2 and 4, and men's beard is removed in column 3 of Figure 4 and column 1 of Figure 5.

We also show results on semantic segmentation in Figure 6 in which we consider non-targeted attacks on DeepLab-v2 [Chen et al. (2017)] with a generator trained on the COCO-stuff dataset [Caesar et al. (2018)]. We iteratively modify modulation parameters at all layers, using a step size of $0.001$, to maximize the segmentation loss with respect to the given label map. As we observe, subtle modifications to images lead to large drops in accuracy.

Unlike perturbation-based attacks, $L_p$ distances between original and adversarial images are large, yet they are visually similar. Moreover, we do not observe high-frequency perturbations in the generated images. The model

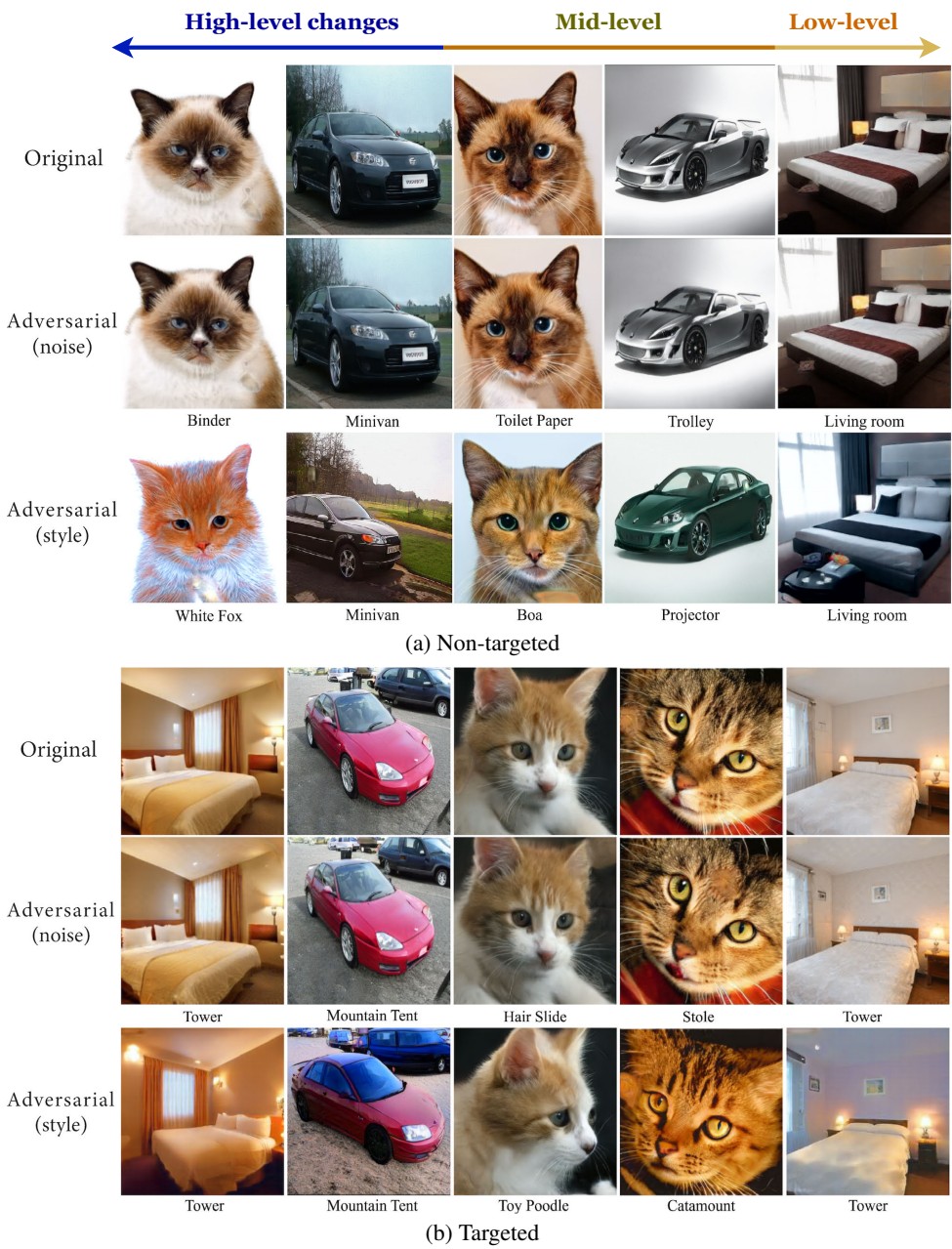

Figure 3: Unrestricted adversarial examples on LSUN for a) non-targeted and b) targeted attacks. Predicted classes are shown under each image.

learns to modify the initial input without leaving the manifold of realistic images. Additional examples and higher-resolution images are provided in the appendix.

## 4.1 ADVERSARIAL TRAINING

Adversarial training increases robustness of models by injecting adversarial examples into training data. Adversarial training with norm-bounded examples degrades performance of the classifier on clean images as they have different underlying distributions. We show that adversarial training with our unrestricted examples *improves* the model's accuracy on clean images. To ensure that the model maximally benefits from these additional samples, we need to avoid unrealistic examples which do not resemble natural images. Therefore, we only include samples that can fool the model in less than a specific number of iterations. We use a threshold of 10 as the maximum number of iterations, and demonstrate results on classification and semantic segmentation. We use the first 10 generated examples for each starting image in the segmentation task. Table 1 shows accuracy of the strengthened and orig-

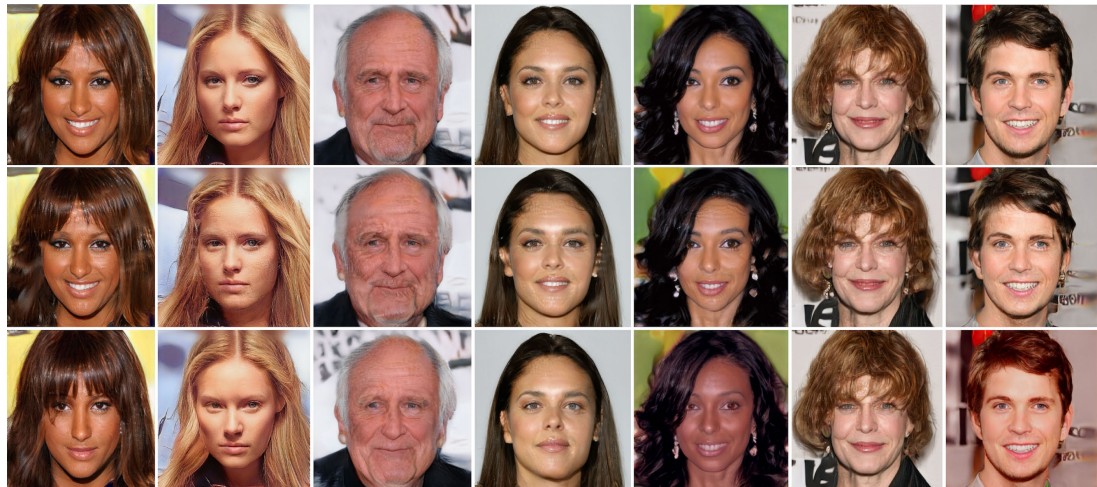

Figure 4: Unrestricted adversarial examples on CelebA-HQ gender classification. From top to bottom: original, noise-based and style-based adversarial images. Males are classified as females and vice versa.

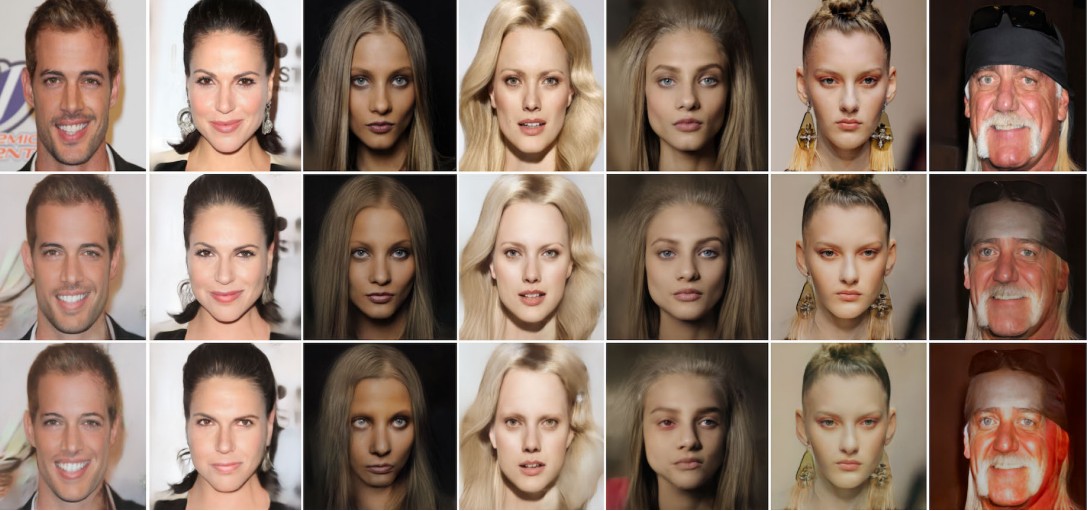

Figure 5: Input-conditioned adversarial examples on CelebA-HQ gender classification. From top to bottom: input, generated and style-based images. Males are classified as females and vice versa.

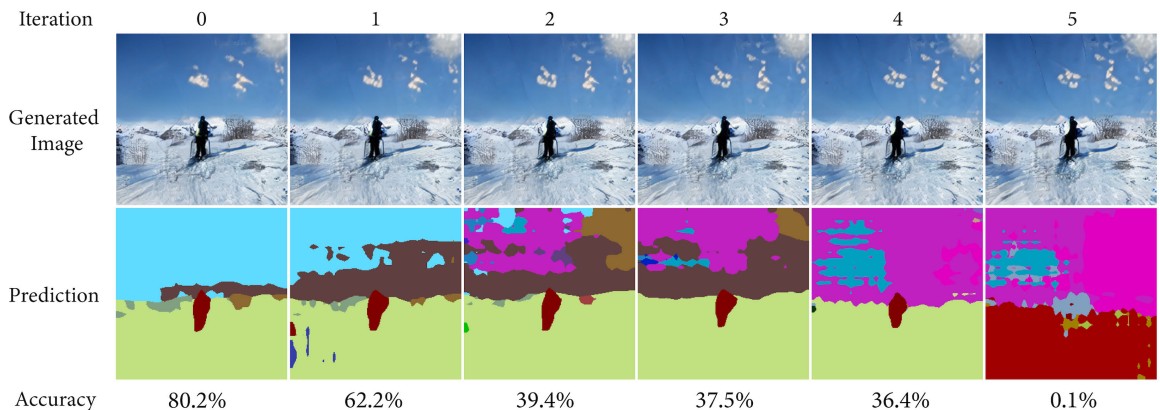

Figure 6: Unrestricted adversarial examples for semantic segmentation. Generated images, corresponding predictions and their accuracy (ratio of correctly predicted pixels) are shown for different number of iterations.

inal classifiers on clean and adversarial test images. For the segmentation task we report the average accuracy of adversarial images at iteration 10. Similar to norm-constrained perturbations, adversarial training is an effective defense against our unrestricted attacks. Note that accuracy of the model on clean test images is improved after adversarial training. This is in contrast to training with norm-bounded adversarial inputs which hurts the classifier's performance on clean images, and it is due to the fact that unlike perturbation-based inputs, our generated images live on the manifold of realistic images as constrained by the generative model.

| | Classification (LSUN) | | Classification (CelebA-HQ) | | Segmentation | |
|---|---|---|---|---|---|---|
| | Clean | Adversarial | Clean | Adversarial | Clean | Adversarial |
| Adv. Trained | **89.5%** | 78.4% | **96.2%** | 83.6% | **69.3%** | 60.2% |
| Original | 88.9% | 0.0% | 95.7% | 0.0% | 67.9% | 2.7% |

Table 1: Accuracy of adversarially trained and original models on clean and adversarial test images.

## 4.2 USER STUDY

Norm-constrained attacks provide visual realism by $L_p$ proximity to a real input. To verify that our unrestricted adversarial examples are realistic and correctly classified by an oracle, we perform human evaluation using Amazon Mechanical Turk. In the first experiment, each adversarial image is assigned to three workers, and their majority vote is considered as the label. The user interface for each worker contains nine images, and shows possible labels to choose from. We use 2400 noise-based and 2400 style-based adversarial images from the LSUN dataset, containing 200 samples from each class (10 scene classes and 2 object classes). The results indicate that 99.2% of workers' majority votes match the ground-truth labels. This number is 98.7% for style-based adversarial examples and 99.7% for noise-based ones. As we observe in Figure 3, noise-based examples do not deviate much from the original image, resulting in easier prediction by a human observer. On the other hand, style-based images show coarser changes, which in a few cases result in unrecognizable images or false predictions by the workers.

We use a similar setup in the second experiment but for classifying real versus fake (generated). We also include 2400 real images as well as 2400 unperturbed images generated by Style-GAN. 74.7% of unperturbed images are labeled by workers as real. This number is 74.3% for noise-based adversarial examples and 70.8% for style-based ones, indicating less than 4% drop compared with unperturbed images generated by Style-GAN.

## 4.3 EVALUATION ON CERTIFIED DEFENSES

Cohen et al. (2019) propose the first certified defense on norm-bounded perturbations at the scale of ImageNet. Using randomized smoothing with Gaussian noise, their defense guarantees a certain top-1 accuracy for perturbations with $L_2$ norm less than a specific threshold. We demonstrate that our unrestricted attacks can break this certified defenses on ImageNet. We use 400 noise-based and 400 style-based adversarial images from the object categories of LSUN, and group all relevant ImageNet classes as the ground-truth. Our adversarial examples are evaluated against a randomized smoothing classifier based on ResNet-50 using Gaussian noise with standard deviation of 0.5. Table 2 shows accuracy of the model on clean and adversarial images. As we observe, the accuracy drops on adversarial inputs, and the certified defense is not effective against our attack. Note that we stop updating adversarial images as soon as the model is fooled. If we keep updating for more iterations afterwards, we can achieve even stronger attacks.

| | Accuracy |
|---|---|
| Clean | 63.1% |
| Adversarial (style) | 21.7% |
| Adversarial (noise) | 37.8% |

Table 2: Accuracy of a certified classifier equipped with randomized smoothing on our adversarial images.

## 5 CONCLUSION AND FUTURE WORK

The area of unrestricted adversarial examples is relatively under-explored. Not being bounded by a norm threshold provides its own pros and cons. It allows us to create a diverse set of attack mechanisms; however, fair comparison of relative strength of these attacks is challenging. It is also unclear how to even define provable defenses. While several papers have attempted to interpret norm-constrained attacks in terms of decision boundaries, there has been less effort in understanding the underlying reasons for models' vulnerabilities to unrestricted attacks. We believe these can be promising directions for future research. We also plan to further explore transferability of our approach for black-box attacks in the future.

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

## A APPENDIX

### A.1 COMPARISON WITH SONG ET AL. (2018)

We show that adversarial training with examples generated by Song et al. (2018) hurts the classifier's performance on clean images. Table 3 demonstrates the results. We use the same classifier architectures as Song et al. (2018) and consider their basic attack. We observe that the test accuracy on clean images drops by $1.3\%$, $1.4\%$ and $1.1\%$ on MNIST, SVHN and CelebA respectively. As we show in Table 1 training with our examples improves the accuracy, demonstrating difference of our approach with that of Song et al. (2018).

| | MNIST | | SVHN | | CelebA | |
|---|---|---|---|---|---|---|
| | Clean | Adversarial | Clean | Adversarial | Clean | Adversarial |
| Adv. Trained | 98.2% | 84.5% | 96.4% | 86.4% | 96.9% | 85.9% |
| Original | 99.5% | 12.8% | 97.8% | 14.9% | 98.0% | 16.2% |

Table 3: Accuracy of adversarially trained and original models on clean and adversarial test images from Song et al. (2018).

To further illustrate and compare distributions of real and adversarial images, we use a pre-trained VGG network to extract features of each image from CelebA-HQ, our adversarial examples, and those of Song et al. (2018), and then plot them with t-SNE embeddings as shown in Figure 7. We can see that the embeddings of CelebA-HQ real and our adversarial images are blended while the those of CelebA-HQ and Song et al.'s adversarial examples are more segregated. This again provides evidence that our adversarial images stay closer to the original manifold and hence could be more useful as adversarial training data.

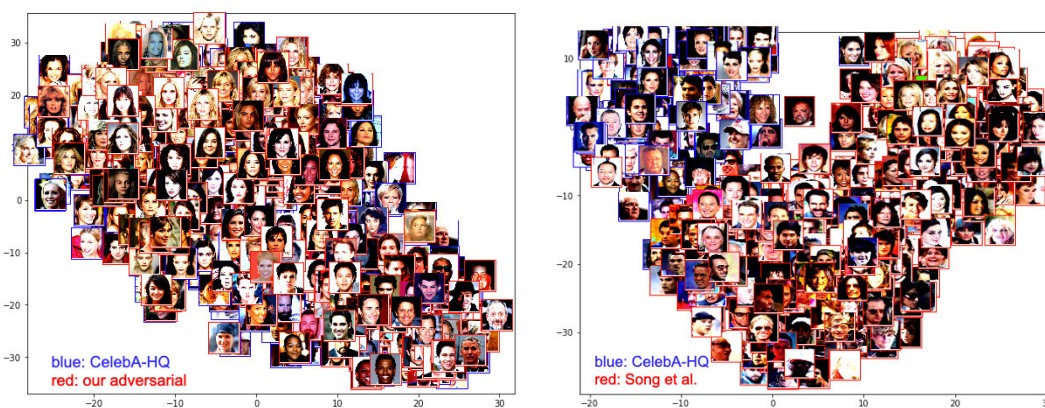

Figure 7: t-SNE plot comparing distributions of real images with adversarial examples from our approach and Song et al.

### A.2 ADVERSARIAL TRAINING WITH NORM-BOUNDED PERTURBATIONS

We consider adversarial training with norm-bounded perturbations and limit the number of iterations to make the setup comparable with our unrestricted adversarial training. Specifically, we use Iterative-FGSM with $\epsilon = 4$ and a bounded number of steps. Results are shown in Table 4. Note that accuracy of the models drop on clean images although we use a weak attack. This is in contrast to training with our unrestricted adversarial examples that improves the accuracy.

### A.3 NUMBER OF ITERATIONS

To make sure the iterative process always converges in a reasonable number of steps, we measure the number of updates required to fool the classifier on 1000 randomly-selected images. Results are shown in Table 5. Note that for targeted attacks we first randomly sample a target class different from the ground-truth label for each image.

|  |  | IFGSM-2 | IFGSM-5 |
|---|---|---|---|
|  | Original | Adv. Trained | Adv. Trained |
| LSUN | 88.9% | 88.4% | 87.8% |
| CelebA-HQ | 95.7% | 95.1% | 94.6% |

Table 4: Adversarial Training with norm-bounded perturbations. Iterative-FGSM ($\epsilon = 4$) with a maximum of 2 and 5 iterations is considered, and accuracy of the adversarially trained and original models on clean test images are shown.

|  | LSUN | | CelebA-HQ |
|---|---|---|---|
|  | Targeted | Non-targeted | |
| Style-based | $9.1 \pm 4.2$ | $6.8 \pm 3.6$ | $7.3 \pm 3.0$ |
| Noise-based | $4.5 \pm 1.7$ | $3.7 \pm 1.8$ | $6.2 \pm 4.1$ |

Table 5: Average number of iterations (mean $\pm$ std) required to fool the classifier.

## A.4 Object Detection Results

Figure 8 illustrates results on the object detection task using the RetinaNet target model [Lin et al. (2017)]. We observe that small changes in the images lead to incorrect bounding boxes and predictions by the model.

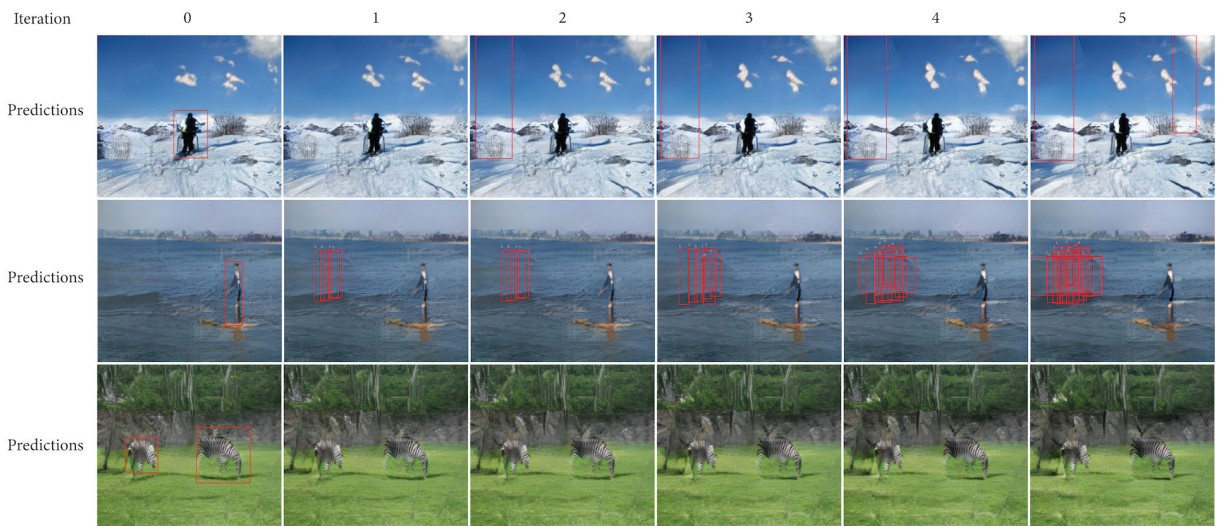

Figure 8: Unrestricted adversarial examples for object detection. Generated images and their corresponding predictions are shown for different number of iterations.

## A.5 Impact of $\gamma$ and $\beta$ on Semantic Segmentation results

In segmentation results shown in Figure 11 we simultaneously modify both $\gamma$ and $\beta$ parameters of the SPADE module (Figure 2). We can also consider the impact of modifying each parameter separately. Figure 9 illustrates the results. As we observe, changing $\gamma$ and $\beta$ modifies fine details of the images which are barely perceptible yet they lead to large changes in predictions of the segmentation model.

## A.6 Adversarial Changes to Single Images

Figure 10 illustrates how images vary as we manipulate specific layers of the network. We observe that each set of layers creates different adversarial changes. For instance, layers 12 to 18 mainly change low-level color details.

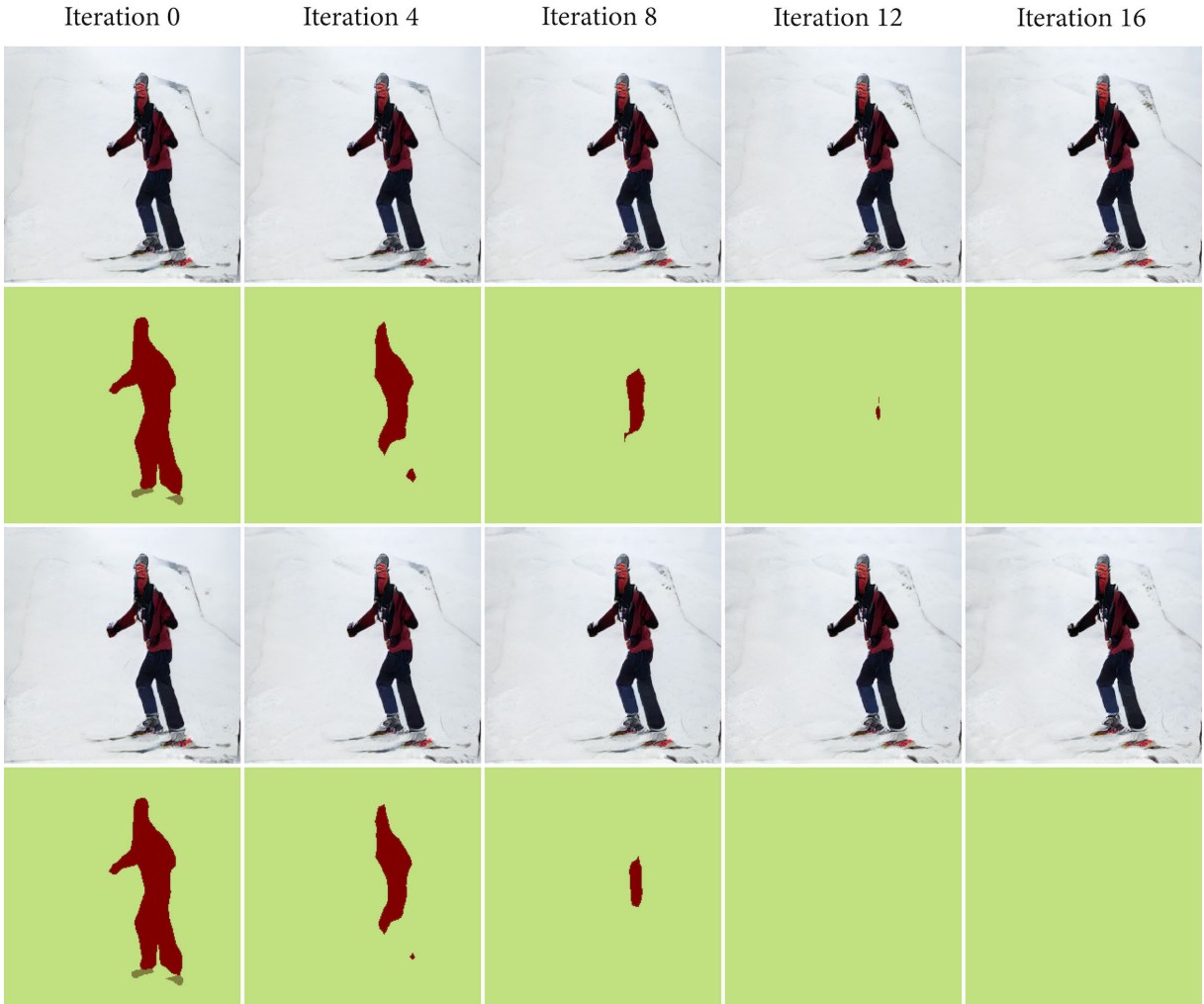

Figure 9: Impact of separately modifying $\gamma$ and $\beta$ parameters on segmentation results. Modified images at different iterations and corresponding predictions are shown. In the first two rows only the $\gamma$ values are changed and in the last two rows only the $\beta$ values are modified.

## A.7 ADDITIONAL EXAMPLES

We also provide additional examples and higher-resolution images in the following. Figure 11 depicts additional examples on the segmentation task. Figure 12 illustrates adversarial examples on CelebA-HQ gender classification, and Figure 13 shows additional examples on the LSUN dataset. Higher-resolution versions for some of the adversarial images are shown in Figure 14, which particularly helps to distinguish subtle differences between original and noise-based images.

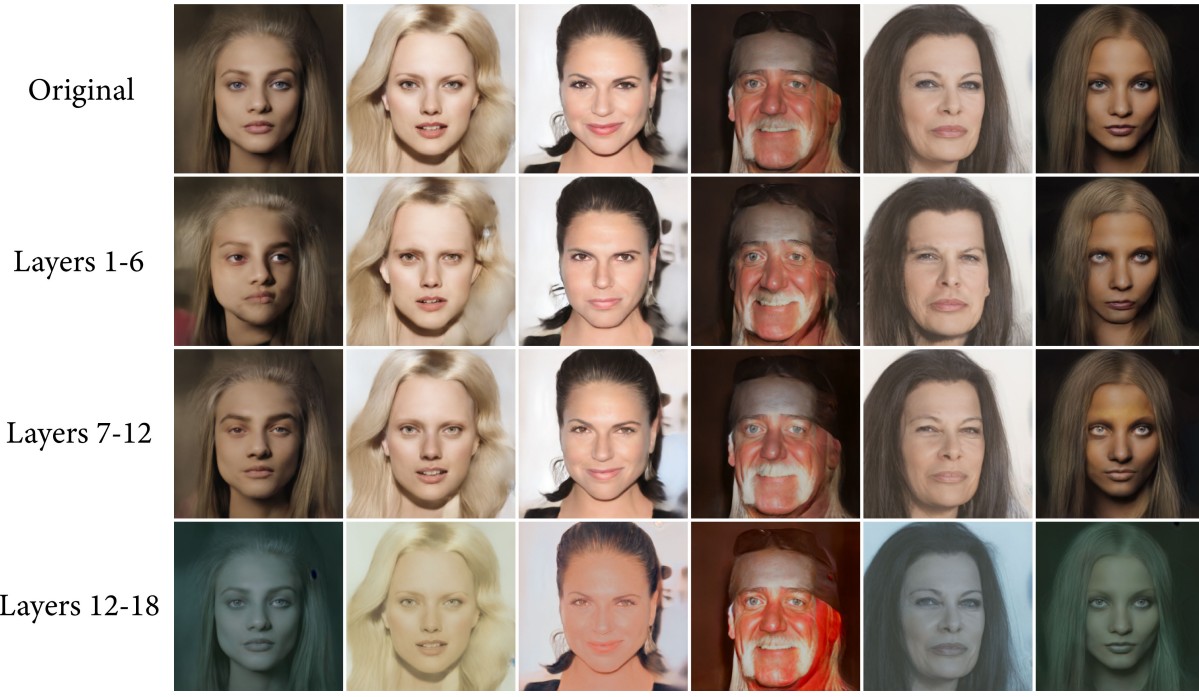

Figure 10: Impact of manipulating different layers of the network on generated adversarial images.

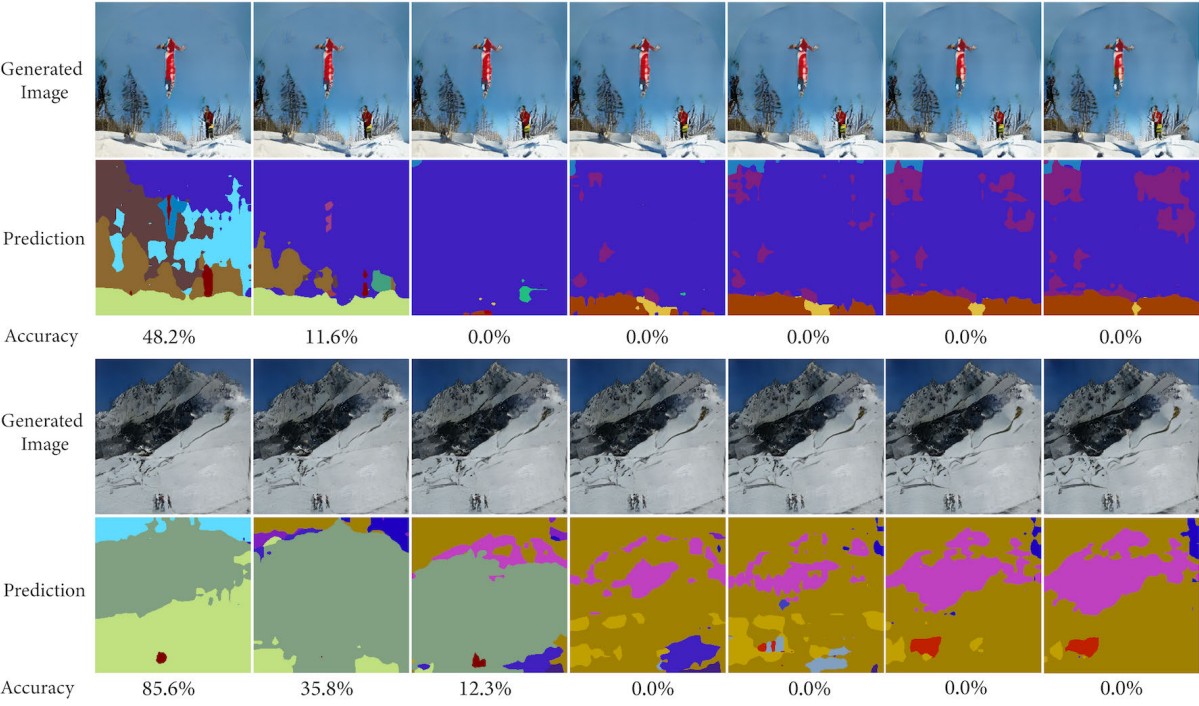

Figure 11: Unrestricted adversarial examples for semantic segmentation. Generated images, corresponding predictions and their accuracy (ratio of correctly predicted pixels) are shown for different number of iterations.

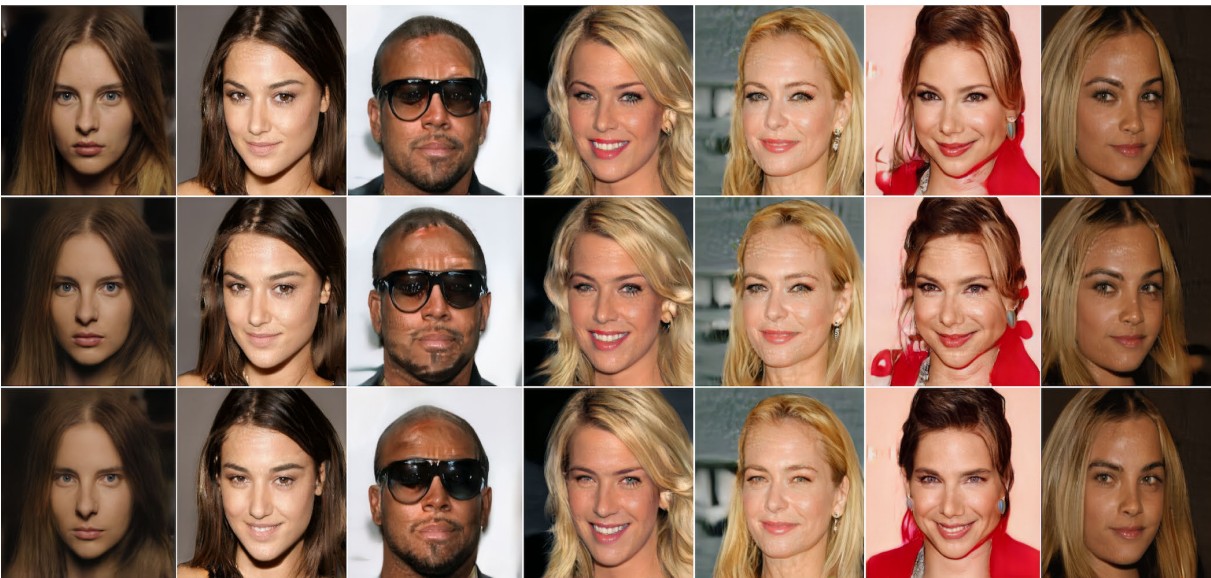

Figure 12: Unrestricted adversarial examples on CelebA-HQ gender classification. From top to bottom: Original, noise-based and style-based adversarial images. Males are classified as females and vice versa.

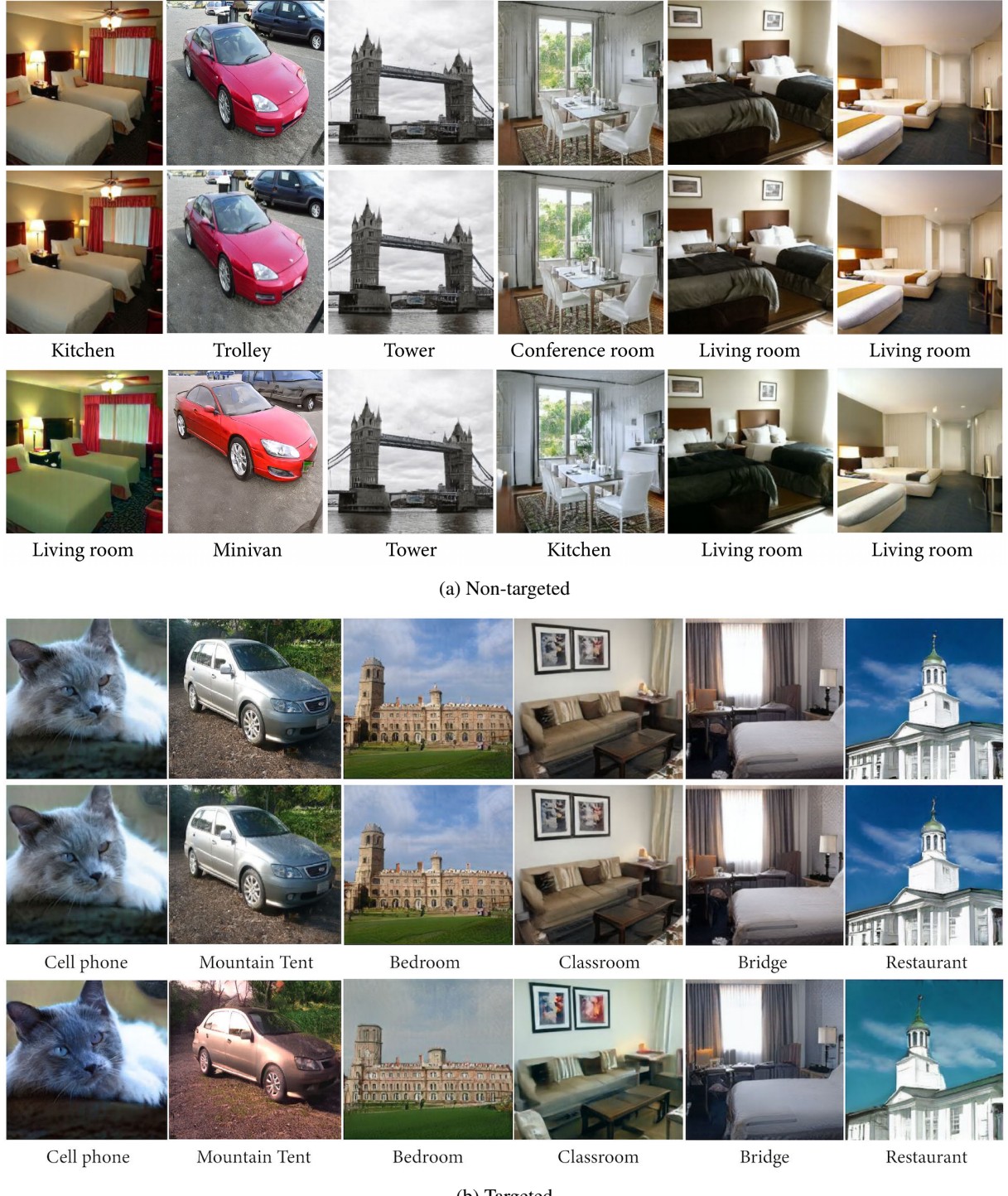

(a) Non-targeted

(b) Targeted

Figure 13: Unrestricted adversarial examples on LSUN for a) non-targeted and b) targeted attacks. From top to bottom: original, noise-based and style-based images.

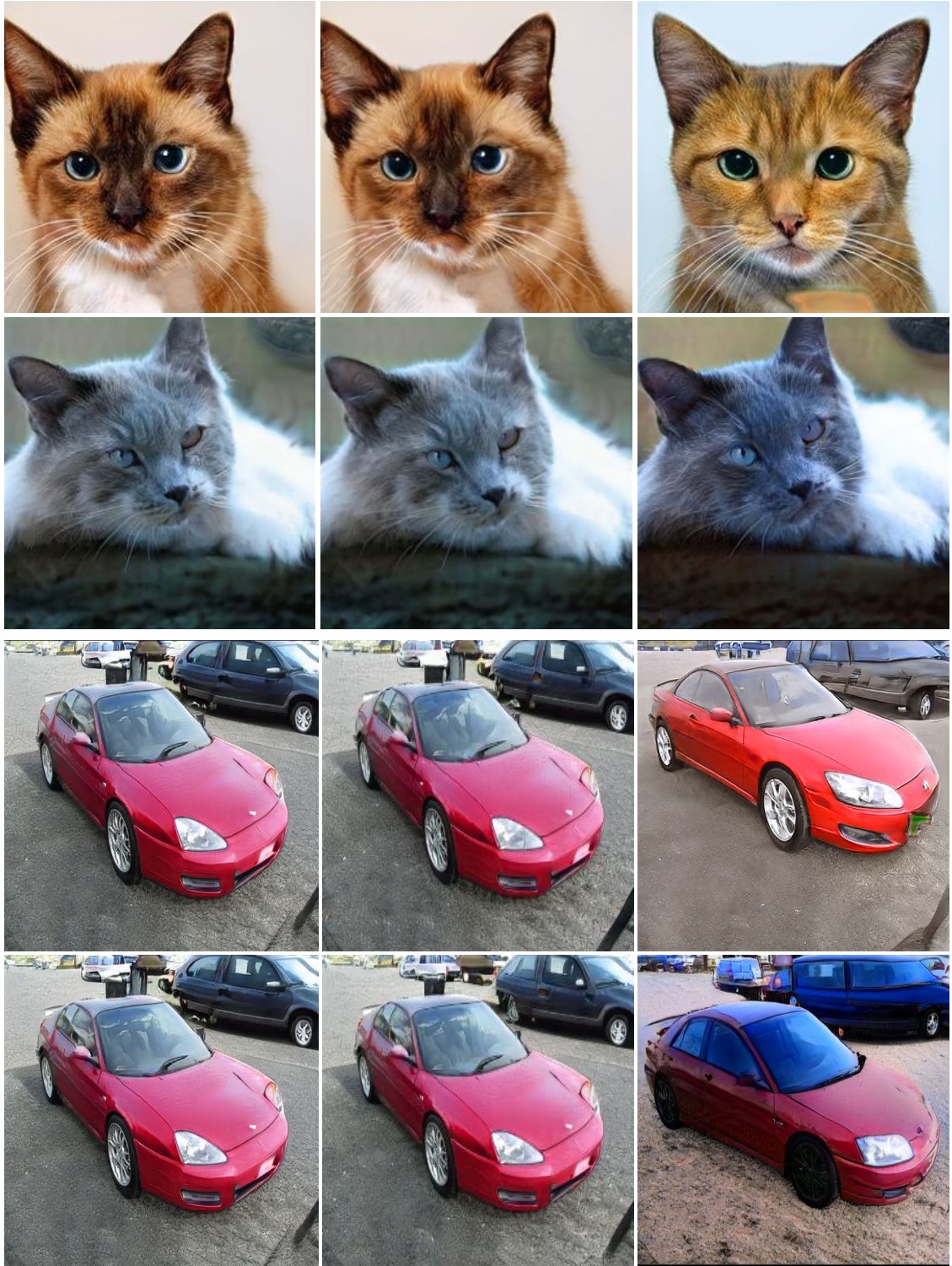

Figure 14: High resolution versions of adversarial images. From left to right: original, noise-based and style-based images.

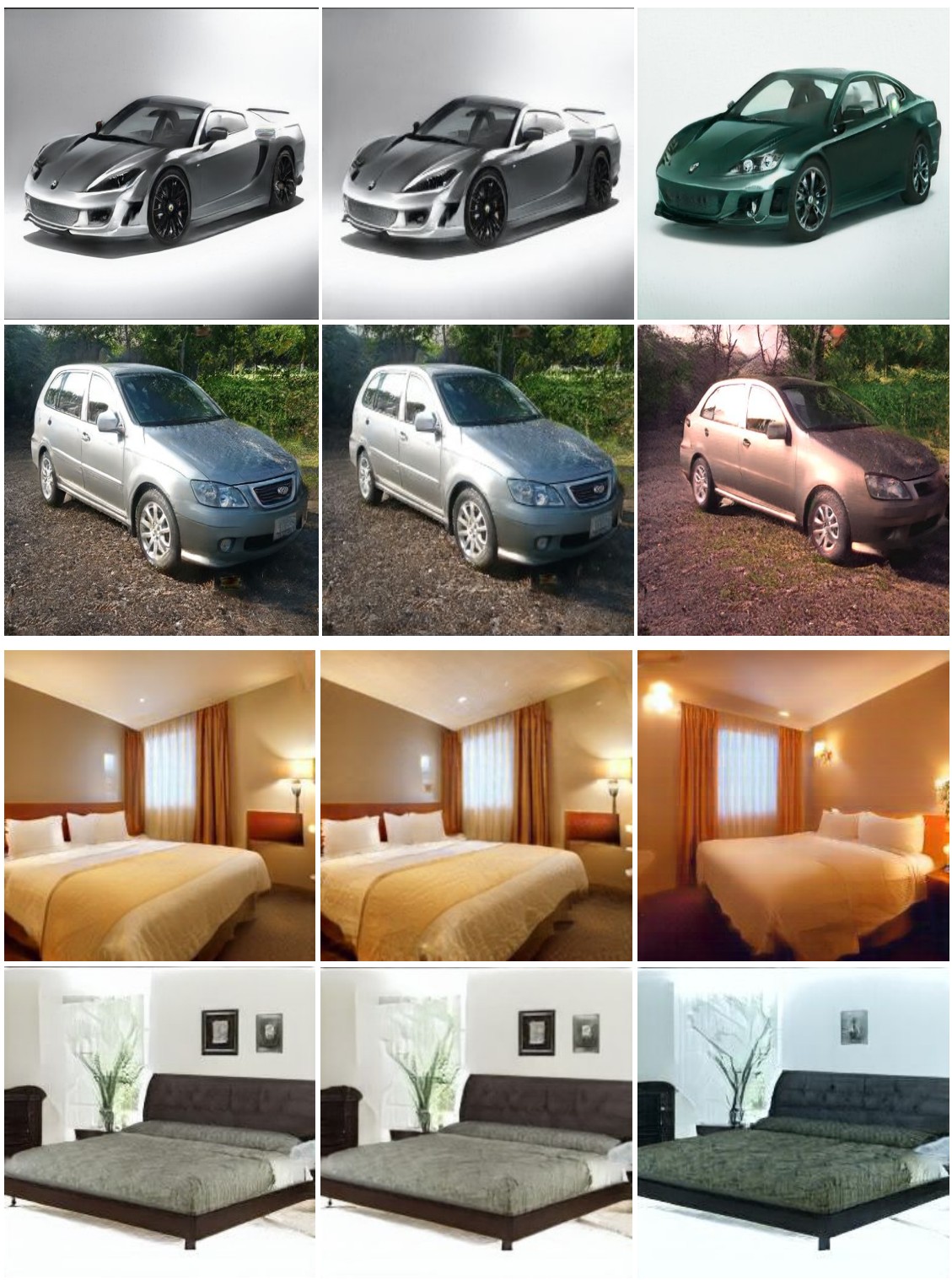

Figure 14: (cont.) High resolution versions of adversarial examples. From left to right: original, noise-based and style-based images.

