# OpenReview forum: "Fine-grained Synthesis of Unrestricted Adversarial Examples"
_ICLR.cc/2021/Conference — Reject_

### Official Review · AnonReviewer2 · 2020-10-28

**Rating:** 7
**Confidence:** 3

**Review:**

###Summary###


The paper proposes a method of generating adversarial samples to enhance classification performance.
Specifically, it finds variables of pre-trained generative model to produce images that the pre-trained classifier gives wrong answers.
The new classifier is then trained by adding these samples to the existing training dataset.
This method achieved good performance because the distributions of the samples generated in this way is closer to the distribution of the real images than the ones generated with norm-bounded perturbations.


###Pros###


-
The effectiveness of the proposed method is reasonably explained by comparing with the preceding works.


-
Authors thoroughly analyzed the qualitative results.
It helps in understanding the underlying mechanisms.



###Questions###


-
Is there a difference in performance between using non-targeted adversarial samples and targeted adversarial samples?
If so, how different is it?
Which of the two to use depends on the outcome?

-
What is the criteria that you divide layers as high-, mid- and low-level ones?
Have you checked layer-wise effect?

-
How one image varies in different settings?
For example, generation results in figure 3 using only one original image.

---

> ### Author Response · Authors · 2020-11-23
> **Response to Reviewer 2**
>
> Thank you for your comments and feedback. We respond to the questions in the following:
>
> - In our attacks we iterate until we fool the classifier, so the fooling rate is 100% for both targeted and non-targeted attacks. In general attackers choose whether they want to only change the model’s original prediction (non-targeted attack) or force the model to predict a specific label (targeted attack). It is easier to create non-targeted adversarial examples as the output can be any label other than the original prediction.
>
> - We use equal splits for dividing layers to high-level (top 6 layers), mid-level (layers 7 to 12) and low-level (bottom 6 layers). However, there is no fixed criterion for this. The StyleGAN paper utilizes top 2 layers for high-level changes, layers 3 to 4 for mid-level changes and other layers for fine details (see Figure 3 of the StyleGAN paper). We can obtain more granular changes by layerwise manipulation, i.e. changing only one of the 18 layers.
>
> - We have added results of varying single images in the appendix (section A.6) considering CelebA-HQ gender classification. We can observe a range of low-level to high-level adversarial changes.

---

### Official Review · AnonReviewer3 · 2020-10-29
**This work proposes a new perturbation method for generating unrestricted adversarial examples through introduction of stylistic and stochastic modifications.**

**Rating:** 6
**Confidence:** 2

**Review:**

The paper presents a new method for generating unrestricted adversarial examples.  Based on Style-GAN, this work separates stylistic and noise modifications so as to control higher-level aspects and lower-level aspects of image generation.

By handling style and noise variables separately and changing the different levels of synthesis networks, the model can input various types of perturbations in generating adversarial images.  As the authors claim, the style variables from different layers affect different aspects of images.  Generation of adversarial images are tested in both un-targeted and targeted attacks.  Overall,
the paper is well-motivated, well-written, and the method is evaluated with three tasks, classification, semantic segmentation, and object detection.

On the other hand, although the different layers of the networks are concerned with different aspects of the images and the proposed method can generate a variety of images, we may not be able to intentionally control specific aspects of the images.  This is an incremental work on top of Style-GAN so that the novelty of the paper is not very high.

Please make clear how the parameter values are determined.  For example, how did you select the step sizes?

---

> ### Author Response · Authors · 2020-11-23
> **Response to Reviewer 3**
>
> Thank you for your comments and feedback. We provide clarifications in the following:
>
> - We consider the following trade-off in choosing the step sizes: very large step sizes can lead to unrealistic artifacts since the latent values can go beyond the bound of values seen during training. Using very small step sizes increases the number of iterations and makes inference and adversarial training slow. In practice, we choose the step sizes such that the algorithm converges in a reasonable number of steps (e.g. average number of iterations to fool the model is less than 10).
>
> - We have general control over different aspects of images through the disentangled latent space. Admittedly, this disentangling is not perfect and as generative models improve, the attacker can have more control over the generation process. Nevertheless, the general algorithm remains the same.

---

### Official Review · AnonReviewer4 · 2020-10-31
**Style-based adversarial attack is on the way**

**Rating:** 6
**Confidence:** 3

**Review:**

This paper proposes a mechanism to generate adversarial examples by applying latent variables level manipulation, based on the styleGAN framework. Unlike previous works mostly focused on image level perturbations and geometry transformations, this work tends to control higher level latent sampling such as style, so as to generate a style-adversarial examples. Although a similar idea has been proposed by Song et al. (2018), this work is along the same direction and achieves better performance. The loss is proposed for general classification tasks such as object classification, object detection and semantic segmentation. The experimental results show not only qualitatively confusing human vision but also quantitatively improve the performance on testing clean images.

+ The paper is well written and easy to read.
+ Experimental results demonstrate the proposed idea qualitatively and quantitatively.
+ Sufficient ablation analysis to make the proposed method convincing.

However, I still have some concerns:

- There’s no experiment to compare with existing methods such as Xie et al. ‘20 and PGD. A standard experimental protocol ex ImageNet should be conducted and fairly compared.
- Deeper analysis of the impact by y_adv and eta_adv in both object detection and semantic segmentation. Although final results in table 1 shows that style-based adversarial training benefits the performance, ablation studies of y_adv and eta_adv in both tasks should be conducted.
- Since the attack is in the feature space, the defense should also happen in the feature space [a];

Overall, I think the paper is valuable. The proposed idea is novel, and sufficient experiments are provided to demonstrate the idea. Rich visualization to help readers understand the concept. I’m willing to raise my rate if my concerns are addressed.

[a] M. Lecuyer et al., Certified Robustness to Adversarial Examples with Differential Privacy: https://arxiv.org/abs/1802.03471

---

> ### Author Response · Authors · 2020-11-23
> **Response to Reviewer 4**
>
> Thank you for your comments and feedback. We address the concerns below:
>
> - PGD is based on norm-bounded perturbations and it is well known that it decreases performance of the model on clean images. Table 8 of Xie et al. reports this performance drop. We have also added an experiment on Iterative-FGSM (which is similar to PGD) in the appendix (section A.2). Our experiment shows that adversarial training with I-FGSM decreases accuracy of the model on clean images while our approach improves the performance.
> Regarding comparisons with Xie et al., note that they only focus on norm-bounded perturbations and utilize auxiliary batch norm layers to prevent degradation of the model on clean images after adversarial training. Our approach focuses on unrestricted adversarial examples and does not require any modifications to the architecture. Both approaches improve performance of the model on clean images and can be potentially combined for further performance gain. Since StyleGAN requires training a separate model for each class, it needs datasets with a large number of samples per class. However, ImageNet only has 1000 images per class, so it is not possible to directly compare the two methods on ImageNet.
>
> - We have added an experiment in the appendix (section A.5) to demonstrate the impact of gamma_adv and beta_adv on segmentation results. We observe that changing each variable results in fine changes which are barely perceptible yet they lead to large changes in predictions of the model.
>
> - Thank you for noting the paper [a]. We have added it to the related work and mentioned that it can be an alternative defense at the ImageNet scale. While our attack is in the feature space, we feed the *images* to the defended classifier so the defense can actually either be in the image space or the feature space, and the experiment on Cohen et al.’s defense is still valid.

---

### Official Review · AnonReviewer5 · 2020-11-03

**Rating:** 4
**Confidence:** 5

**Review:**

Pros:
- Targets an important problem, adversarial attacks semantically constrained as opposed to being constrained by an artificial norm ball.
- Extensive results with a wide variety of models, datasets, and more importantly, applications, with not only attack evaluation on standard models, but application to adversarial training, a user study, and evaluating against certified defenses.

Cons:
- The disentangled representations of StyleGAN were used for generating realistic perturbations, and the application of training with said perturbations (adversarial training) was considered [1]. This paper isn't cited, let alone compared to given the significant similarity in the methodology. [2] considered constraining adversarials to be within the output space of a learned generative model, this work was also not cited.
- The emphasis given to the result that adversarial training improves clean performance isn't fully justified, as unrealistic images, e.g. images not too perturbed, were controlled for by limiting the number of iterations. If one limits the number of iterations for a norm-based attack, or considers a smaller epsilon ball, the same control would be met, but a comparison to norm-based adversarials with the same control is not conducted. Note that adversarial training was originally viewed and used as a regularization method [3,4], improving i,i,d, performance, and the robustness-accuracy tradeoff can straightforwardly be mitigated by considering a weaker form of robustness by using a weaker attack (FGSM [3]), or controlling the strength of the perturbation, where the latter is exactly the filtering performed here. It is intuitive that semantic adversarials would provide benefit for the robustness-accuracy tradeoff, but this experiment certainly does not demonstrate that due to the obvious confounders.
- The emphasis on the "breaking" of a certified defense can also be seen as overclaiming, as breaking a certified defense implies succeeding against the defense within their considered threat model. This is not done here, thus it doesn't violate any of the claims made by randomized smoothing, thus it does not break the certified defense. Though the impracticality of norm-based adversarial defenses are well-understood, this experiment simply shows that if one goes outside of a certified defense's threat model, the certified defense can perform worse, which is entirely expected.

Conclusion: On one hand, the research direction is valuable and the experimental results are extensive. On the other hand, comparison to the literature is sorely lacking, there exists techniques which share much of the same functionality of the method [1], a comparison to norm-based adversarials in the adversarial training experiment should have been done to clarify concerns that controlling for unrealistic adversarials is the source of the result, not the contribution of semantic adversarials, and the certified defense is not "broken", as the attack went outside of the defense's threat model.

In summary, this work does not compare to any baselines when baseline experimentation is clearly needed to justify the novelty and contribution of the work, and fair treatment of prior literature is missing, with not only citations missing for quite similar approaches, but claims are made which on the surface invalidate previous work (breaking randomized smoothing) when said previous work was not fairly evaluated (as the attack went outside of the defense's clearly specified threat model, thus not invalidating any of their claims).

I implore the authors to consider their contribution in the context of the literature more carefully.

References:
 [1]: https://openaccess.thecvf.com/content_CVPR_2020/html/Gowal_Achieving_Robustness_in_the_Wild_via_Adversarial_Mixing_With_Disentangled_CVPR_2020_paper.html.
 [2]: https://openaccess.thecvf.com/content_CVPR_2019/html/Stutz_Disentangling_Adversarial_Robustness_and_Generalization_CVPR_2019_paper.html.
 [3]: https://arxiv.org/abs/1412.6572.
 [4]: https://arxiv.org/abs/1704.03976

---

> ### Author Response · Authors · 2020-11-23
> **Response to Reviewer 5**
>
> Thank you for your comments and feedback. We address the concerns below:
>
> - Thank you for letting us know about [1] and [2]. We have added them to the revised version. Since authors of the recent work [1] have not provided their code, we are not able to directly compare our approach with them but we have cited the paper and described the differences in the related work section (2.2) of our revised manuscript. We provide a more detailed description here: In terms of methodology, adversarial training proposed in [1] differs from us in that they require precomputing a mapping from the image space to the latent space for the whole dataset, which is computationally prohibitive for large datasets. Their approach is also constrained to fine changes using only a subset of the latent variables. They argue that coarse changes might be label-dependent. This statement is not true since coarse stylistic changes should not alter the label (e.g. gender) and merely modify high-level aspects of images. Moreover, [1] only considers the classification task on low-resolution datasets such as ColorMNIST (28x28) and CelebA (64x64). While their approach uses the StyleGAN model, they do not show any results on high-resolution datasets that StyleGAN is originally trained on (e.g. LSUN and CelebA-HQ (1024x1024)), which makes it hard to ascertain that their adversarial training will be effective on high-resolution datasets. Even on low-resolution datasets such as Color-MNIST, their adversarial training can perform worse than random sampling on unbiased datasets as shown in Table 2 of [1].
> On the other hand, our approach directly samples and manipulates latent variables without requiring the mapping step. While [1] builds its methodology upon the notion of label-dependent and label-independent latent variables, we show that this separation is not necessary in practice for the StyleGAN model. We demonstrate that by limiting the number of iterations we can use both coarse and fine changes and both contribute to improvements in performance. Our approach is effective on high-resolution datasets such as CelebA-HQ and LSUN and uses a range of low-level to high-level changes for adversarial training. In addition, we propose the first method for unrestricted adversarial attacks on semantic segmentation and object detection, and demonstrate that adversarial training improves segmentation results on clean images. We show that our adversarial examples can break certified defenses on norm-constrained perturbations and are realistic as verified by human evaluation. Also, please note that [1] is published recently and is similar in essence to Song et al. with which we have already provided a detailed comparison in the appendix (section A.1). We have also cited [2] in the revised version of the paper. [2] is also similar to Song et al., does not consider fine-grained generation and only considers low-resolution datasets such as MNIST and Fashion-MNIST.
>
> - We have added an experiment on adversarial training with controlled norm-bounded perturbations in the appendix (section A.2). We use Iterative-FGSM with epsilon=4 and a bounded number of steps (2 and 5). The results show that adversarial training with weak norm-bounded perturbations decreases accuracy of the models on clean images. With a comparable number of steps, our method not only did not cause degradation in the clean images, it even boosted performance on them.
>
> - We have revised the manuscript (in contributions and section 4.3) to clarify this concern. We specifically note that the certified defense is on norm-bounded perturbations. We did not claim that our attack is within the defense’s specified threat model. We rather wanted to encourage research towards more general defenses that also consider unrestricted attacks.

---

### Author Response · Authors · 2020-11-23
**Overall Response to Reviewers**

We thank all the reviewers for their valuable feedback. Overall, we find that the reviewers agree that our results are thorough and extensive, demonstrate the proposed idea well qualitatively and quantitatively, and sufficient ablation analysis is provided to make the proposed method convincing. They also agree that the paper is well-written and easy to follow. Reviewer 5 notes that the paper targets an important problem and Reviewer 2 mentions that the effectiveness of our method is reasonably explained by comparing it with the preceding works.

The critical feedback from the reviewers identified a number of points in the paper that could be clarified or backed up with additional experimental results. Addressing these issues have substantially improved our paper. Below we respond to individual concerns with clarifications and additional results, which are added to the revised version of the paper.

---

### Decision · Program_Chairs · 2021-01-07
**Final Decision**

**Decision:**

Reject

**Comment:**

The paper received two borderline accept recommendations and one accept recommendation from three reviewers with low confidence and a reject recommendation from an expert reviewer.

Although all reviewers found that the paper addresses an important and challenging problem of semantically constraining adversarial attacks as opposed to constraining them artificially by an artificial norm ball. However, during the discussion phase it has been pointed out that there were some important weaknesses indicating that the paper may need one more evaluation round.  The meta reviewer recommends rejection based on the following observations.

In terms of evaluation, while it is understandable the authors were unable to compare to Gowal et al. due to the lack of publicly available implementation, showing Song et al.'s adversarials hurt performance and and are farther than the image manifold has been found puzzling, as this was done by Song et al. only to keep human prediction the same while changing model prediction. Furthermore, the paper did not contain a user study similar to Song et al. for a fair comparison Finally, the discussion revealed that the comparison to "norm-bounded adversarial inputs" may not have clarified whether this experiment faithfully demonstrates an advantage for the contribution as the norm could be contained to a point where accuracy is not reduced, and the discussion on the certified defense being "broken" was inconclusive.